# Changes in the Amount and Distribution of Soil Nutrients and Neighbours Have Differential Impacts on Root and Shoot Architecture in Wheat (*Triticum aestivum*)

**DOI:** 10.3390/plants12132527

**Published:** 2023-07-02

**Authors:** Habba F. Mahal, Tianna Barber-Cross, Charlotte Brown, Dean Spaner, James F. Cahill

**Affiliations:** 1Department of Biological Sciences, University of Alberta, Edmonton, AB T6G 2E9, Canada; habbamahal@gmail.com (H.F.M.);; 2Département de Biologie, Université de Sherbrooke, Sherbrooke, QC J1K 2R1, Canada; 3Department of Agricultural, Food and Nutritional Science, University of Alberta, Edmonton, AB T6G 2P5, Canada

**Keywords:** *Triticum aestivum*, intraspecific competition, kin recognition, nutrient addition, reproductive effort, fitness, aboveground, crown shyness, belowground, root foraging

## Abstract

Plants exhibit differential behaviours through changes in biomass development and distribution in response to environmental cues, which may impact crops uniquely. We conducted a mesocosm experiment in pots to determine the root and shoot behavioural responses of wheat, *T. aestivum.* Plants were grown in homogeneous or heterogeneous and heavily or lightly fertilized soil, and alone or with a neighbour of the same or different genetic identity (cultivars: CDC Titanium, Carberry, Glenn, Go Early, and Lillian). Contrary to predictions, wheat did not alter relative reproductive effort in the presence of neighbours, more nutrients, or homogenous soil. Above and below ground, the plants’ tendency to use potentially shared space exhibited high levels of plasticity. Above ground, they generally avoided shared, central aerial space when grown with neighbours. Unexpectedly, nutrient amount and distribution also impacted shoots; plants that grew in fertile or homogenous environments increased shared space use. Below ground, plants grown with related neighbours indicated no difference in neighbour avoidance. Those in homogenous soil produced relatively even roots, and plants in heterogeneous treatments produced more roots in nutrient patches. Additionally, less fertile soil resulted in pot-level decreases in root foraging precision. Our findings illustrate that explicit coordination between above- and belowground biomass in wheat may not exist.

## 1. Introduction

Resource capture and plant–plant social interactions are inherently spatially explicit, driven by specific plant organs placed in specific locations at specific points in time. Further, the location of specific organs can influence immediate and future resource capture [1,2,3]. Thus, there has likely been powerful selection pressures on plants to exhibit high levels of plasticity in organ placement in response to local conditions, a phenomenon falling within the larger domain of plant behavioural ecology [4,5,6,7]. Accordingly, current research suggests that plants use environmental information to inform behavioural responses that impact their fitness. For example, plants may react to neighbours’ indirect effects, including shading and resource depletion [6]. The behavioural reactions of plants to spatially explicit features of landscapes involve localized movement and changes in overall development and growth [8,9,10]. Commonly described behaviours include alterations in biomass development and spatial distribution, including differential root production and stem/leaf orientation shifts in response to local conditions [11,12,13,14]. Most studies, however, investigate these behaviours in isolation rather than taking an integrated approach exploring multiple behavioural responses within an individual or population. Thus, whether plants integrate information about stimuli and exhibit overall coordination in these above- or belowground responses, or whether there are differences in shoot and root responses to the same stimuli, remains unknown.

A plant’s responses above and below ground due to other organisms and their placement in the environment may result in altered resource capture [1,15,16,17,18]. These responses have cascading effects on neighbouring plants since competition is a prevalent ecological and evolutionary driver [17]. For example, plants may exhibit crown shyness [19,20,21,22], altering shoot placement to reduce leaf overlap in response to neighbouring plants within a potentially shared canopy space. Such a behavioural response alters net photosynthesis [23,24], potentially impacting plant growth and fitness. Behavioural responses to neighbour shoots are not limited to forested systems, and there could be fitness benefits in any system with moderate to dense plant communities. For example, individuals in dense habitats such as crop fields could be victims of density-dependent mortality, but they may prevent this by shifting their crowns strategically [24,25]. Within root systems, plants may move root biomass depending on nutrient availability, soil patchiness, and the presence and identity of neighbouring plants [26,27,28]. Although these behaviours are well supported by ecological and behavioural theory, it is also well established that in animal species analogous behaviours are context-dependent [29,30,31,32]. However, this contextual information is lacking for most plant species, particularly non-model species.

Evolutionary theory suggests that individuals may alter competition with neighbours under certain conditions as a function of their degree of relatedness [33,34]. There is mixed support for kin-dependent behavioural responses in plants [35,36], and these kin-dependent effects may be locally contingent upon the relative costs or benefits of different behavioural decisions. Kin selection theory ties together inclusive fitness and altruism between close relatives in animal and plant species, which is highly important for crop plants, typically genetic monocultures. Many plant species have developed mechanisms to detect and determine both the neighbours’ presence and identity, including changes in light quality or shading [37], touch [38,39], and volatile organic compounds [22,40,41]. Plants then alter aboveground biomass to facilitate or hinder neighbour development [22]. Accordingly, with plant reproduction, studies have shown that some species might have higher reproductive success when grown with siblings than with unrelated plants [7,21,42], some have decreased seed yields [43,44,45,46,47], and some exhibit no difference in reproduction [45]. Similarly, many wild and domesticated species can distinguish between neighbours below ground, indicating kin versus non-kin [22,48] and self-recognition versus non-self-recognition [49,50,51]. Studies looking at root-mediated kin recognition show shifts in biomass allocation [52,53,54,55], lateral root density [55], root branching intensity [55], specific root length [56], and resource uptake rate [20,57]. These species may engage in a tragedy of the commons, allocating more resources to root development to maximize nutrient capture at the expense of their competitors [49,58,59,60,61]. This is highlighted in studies focusing on root foraging precision. Plants exhibit plasticity in root development and spatial distribution when dealing with the interplay of nutrient density or neighbour presence [18,22,62,63,64], typically producing more roots in high-nutrient, high-yield soil patches [49,59,65,66,67].

However, whether plants coordinate above- and belowground behavioural responses to nutrients and neighbours or whether root and shoot systems respond individually to local conditions remains unknown. Studies on plant behaviours in patchy environments have been concerned with changes in a single plant’s above- or belowground biomass distribution, leaving significant literature gaps. Accordingly, direct connections between changes below ground and the aboveground repercussions remain unclear. Finally, no studies have examined the combined effects of intraspecific kin recognition, soil fertility, and nutrient placement both above and below ground. Our study explored the coordination between shoot and root behavioural responses in wheat (*Triticum aestivum* L.). We used wheat because of its global importance as a food crop, and though many studies have investigated the morphological traits and yield of different cultivars [68,69,70], few have focused on behaviour. Furthermore, though there is evidence that wheat alters root and shoot proliferation in the presence of neighbours [61,71] and resource patches [72,73,74], no studies have explored the effects of both factors simultaneously, making it an ideal choice.

We conducted a mesocosm experiment on five different wheat varieties where we manipulated soil nutrient amounts, soil heterogeneity, and neighbour genotypic identity. We hypothesized that *Triticum aestivum* L. plants, in the presence of a related neighbour, would have increased reproductive effort and reproductive biomass due to kin-related behavioural adjustments. Furthermore, when looking at kin effects above ground, we hypothesized that plants grown with neighbours would exhibit increased neighbour avoidance and crown shyness, decreasing the use of the shared aerial space between the plants. Consequently, plants would decrease crown shyness and aboveground avoidance in highly nutritious or homogenous soil as the cost of using shared aerial space is likely lessened due to potentially increased aboveground biomass and overall photosynthetic capabilities. Finally, in studying kin effects below ground, we predicted plants would exhibit increased neighbour avoidance in the presence of related plants, decreasing their foraging precision in nutrient patches and the directly shared soil space to mitigate competition. Accordingly, the value of the nutrient patch is likely greater at low soil fertility levels, resulting in increased root distribution in the nutrient patches due to increased root foraging efforts by the plant.

## 2. Results

### 2.1. Fitness Effects

We found that *Triticum aestivum* L. plants generally had high levels of reproductive effort (Figure 1). However, it was slightly higher when plants were grown alone than with a neighbour, regardless of the neighbour’s identity, soil fertility, or nutrient distribution (df1, df2 = 2, 583, F = 2.758, *p* = 0.0643, Table 1). Additionally, contrary to our hypothesis, plants grown in lower nutrient conditions illustrated higher levels of reproductive effort than those grown in higher nutrient conditions, irrespective of nutrient distribution (df1, df2 = 1, 583, F = 7.243, *p* =0.0073, Table 1). However, there was no difference in the reproductive effort between plants grown in homogeneous or patchy soil (df1, df2 = 1, 583, F = 0.66, *p* = 0.4169, Table 1). For the overall interaction between nutrient level and soil homogeneity, there were no significant differences between plants grown alone or with neighbours (df1, df2 = 2, 583, F = 0.271, *p* = 0.7628, Table 1).

### 2.2. Aboveground Biomass Distribution

We did not observe a significant overall effect of neighbours (df1, df2 = 2, 583, F = 1.522, *p* = 0.219, Table 1, Figure 2). However, plants grown with neighbours exhibited slightly increased shoot asymmetry under high-fertility soil conditions, regardless of neighbour identity (df1, df2 = 2, 583, F = 2.758, *p* = 0.0643, Table 1). Similarly, under low-fertility conditions, shoot asymmetry was slightly lower when plants were grown alone than with neighbours, irrespective of whether the neighbour was kin or stranger (df1, df2 = 2, 583, F = 2.758, *p* = 0.0643, Table 1). Confirming our hypotheses, plants that grew in fertile environments showed significantly higher levels of growth towards the neighbour and central shared area than away (df1, df2 = 1, 583, F = 7.243, *p* = 0.0073, Table 1). When looking at the interactions between the fixed factors, there was no overall impact of neighbour presence, neighbour identity, and soil treatment (df1, df2 = 2, 583, F = 0.271, *p* = 0.7628, Table 1).

### 2.3. Belowground Biomass Distribution

We found that root asymmetry was highly contingent on the nutrient level and soil heterogeneity but not neighbour presence or identity, contrary to our hypothesis (df1, df2 = 2, 582, F = 0.152, *p* = 0.8589, Table 1, Figure 2). However, we did find a significant effect of soil heterogeneity on root asymmetry, depending on the nutrient treatment. Root asymmetry was higher in high-nutrient pots with heterogeneous soil than in low-nutrient, heterogeneous soil (df1, df2 = 582, t = 3.301, *p* = 0.001, Table 2). When overall soil fertility was low, root asymmetry was higher in homogeneous soil than in heterogeneous soil (df1, df2 = 582, t = −2.931, *p* = 0.0035, Table 2). However, when soil fertility was high, there was no discernible difference in root asymmetry between homogeneous and heterogeneous soil treatments (df1, df2 = 582, t = 1.952, *p* = 0.0514, Table 2).

There was no significant effect of neighbour presence and soil treatment interaction on root precision (df1, df2 = 2, 582, F = 0.1577, *p* = 0.562, Table 1, Figure 3). However, we did find increased patch use when the soil was highly nutritious, regardless of nutrient distribution, indicating a lack of interaction (df1, df2 = 1, 582, F = 10.739, *p* = 0.0011, Table 1). Similarly, root biomass in the patches was higher in heterogeneous soil, irrespective of soil fertility (df1, df2 = 1, 582, F = 330.789, *p* ≤ 0.0001, Table 1).

## 3. Discussion

Our results indicate stark differences in wheat’s above- and belowground behavioural responses to neighbours and nutrients. The reproductive effort had only slightly significant kin effects, with reproductive effort marginally higher for plants grown alone in low-nutrient conditions or with neighbours in high-nutrient soil (Figure 1). When looking at the overall aboveground biomass distribution changes, it is apparent that soil fertility, nutrient level, and neighbour presence have a significant impact (Figure 2). Finally, root distribution was affected by the nutrient level, soil heterogeneity, and neighbour presence, with changes in shared soil areas and patch use (Figure 3 and Figure 4).

### 3.1. Reproductive Effort

For reproductive effort, we found that none of the soil structure elements or social interactions we investigated—including soil fertility, nutrient homogeneity, neighbour presence, and neighbour identity—had a drastic impact (Figure 1). Li et al. found that yield was stable across various plant densities when studying maize [47,75,76]. However, output linearly declined when the plant density was above an optimum level set by the species [75]. Over time, the extensive breeding of crop plants may have caused the lack of distinct impact of soil structure and social interactions on the reproductive effort. Crop plants are often bred for traits emphasizing group fitness over individual performance [18,77,78,79,80]. Hence, past selection for inclusive fitness may have favoured more cooperative plant genotypes with features such as shorter stems, erect leaves, and restrained roots [81]. Overall fitness is improved when the negative consequences of competition between kin, including clones, full-siblings, and half-siblings, are minimized [43]. So, the ability of crop species to recognize kin may increase yield by reducing competitive effects [48,81,82,83]. Accordingly, in a study on rice, the data showed that cultivars with kin recognition in mixed cultures increased grain yields, but interestingly, not all cultivars possessed this ability [83]. Furthermore, numerous studies have shown that cooperation based on kin may decrease the prevalence of competitive traits [84,85,86]. Cooperation also allowed for optimized above- and belowground resource capture [35] and subsequent increases in overall fitness [7,87,88].

However, Kiers and Denison refute the notion of crop plants’ emphasis on group fitness, stating that high genetic relatedness, particularly siblings or clones, does not necessarily select for cooperation [81]. Furthermore, single-genotype crop fields would not necessarily provide more significant reproductive effort or yield [82,89]. For example, *Lupinus angustifolius* plants produce significantly more flowers and seeds when grown with unrelated neighbours than with siblings [44]. These differences in kinship effects could result from specific biotic or abiotic environments [45,90]. Our study did not show this above or below ground, indicating that the effect of the relatedness between neighbours is difficult to predict, especially regarding crop species. We lack conclusive evidence of the impacts of kin recognition.

### 3.2. Aboveground Biomass Distribution

Above ground, our results show that the wheat plants increased growth towards the centre of the pot when a neighbouring plant was absent (Figure 2). Some studies have suggested that plants may over-proliferate shoots when grown with a neighbour [49]. Many plant species interpret environmental cues such as shading, volatile organic compounds, touch, and root exudates to determine whether a neighbour is present [22]. Over-proliferation would prove wasteful, however, if the neighbour responds similarly. They would also create a tragedy of the commons, collectively exhausting the resources [1,91]. In our study, the unconstrained growth due to a lack of neighbours enables solitary plants to organize their development solely concerning resource availability, modelling Ideal Free Distribution (IFD) [22].

Our study did not show any distinct impact of neighbour identity on the distribution of aboveground biomass (Figure 2). Wheat may consider all the neighbours as related rather than segregating them into stranger and kin categories if the ‘stranger’ neighbours are not genetically distant enough. This could be attributed to similar genetic backgrounds and close genetic relatedness within the crop [82,83]. In our study, the soil’s nutrient level significantly impacted shoot asymmetry, with the plants increasing neighbour avoidance under low-nutrient conditions (Figure 2). The plants may follow IFD under low-nutrient conditions, minimizing overlap of shared aerial space when belowground resources are already limited but abandoning IFD when belowground resources are plentiful enough for the plants to compete aggressively. The decrease in shading at low nutrient levels could be due to decreased chlorophyll requirements when plants must conserve nutrients due to limited availability. A greenhouse pot experiment on wheat showed that decreasing fertilizer reduced chlorophyll content in the leaves [92]. With a greater need to place roots below ground due to an enhanced necessity for foraging and structural integrity considering the competition, plants would be far less free-handed with allocating resources towards aboveground competition. In some plant species, increasing soil fertility under homogeneous conditions decreased aboveground size-asymmetric competition [93]. Alternatively, increasing nutrient levels in the soil can also inadvertently increase size-asymmetric competition by prompting shoot growth, effectively altering the competition to above ground from below ground [94,95,96], which is more akin to what we observed in our study (Figure 2). Hence, it is apparent from our research that aboveground architecture changes in direct response to alterations in belowground environmental conditions.

### 3.3. Belowground Biomass Distribution and Patch Use

Our study of belowground plant behaviour indicates that soil heterogeneity and nutrient level affected root distribution and patch use (Figure 3 and Figure 4). We saw even root distribution in homogeneous soil treatments, while heterogeneous treatments elicited significant increases in patch use (Figure 4). In the heterogeneous treatments, the plants decreased root growth in all the core locations without a nutrient patch, likely reallocating resources to growth proliferation within the patches, especially in highly nutritive soil. Hence, we observed a cascading effect throughout the pot where the plants use environmental information to optimize root foraging behaviour [66,97,98,99,100,101]. A trade-off exists between maximizing resource intake from the high-quality soil patch and prolonged exploration.

Our study, however, does not indicate any direct effects of neighbour presence or identity on belowground biomass placement (Figure 3). It is likely that wheat either cannot recognize kin or has been bred to disregard familial connections when placing roots below ground. The lack of over-proliferation in the presence of neighbours, especially strangers, has been seen in prior studies [5,102,103,104,105]. However, we did observe the effects of the interaction between nutrient heterogeneity in the soil and neighbour presence. The plants allocated more roots opposite the nutrient patch when they were not facing competition for resources in a limited space with a proximate neighbour (Figure 4). When dealing with a predominately low-nutrient environment with a singular high-nutrient patch equidistant to a neighbour, the plant will expend energy in maximizing resource capture, engaging in a tragedy of the commons. When the plants are free of these restraints and are alone in homogenous soil, they are free to explore the entirety of the soil.

Interestingly, the interaction of nutrient level and soil heterogeneity impacted root growth between the plants and the edge of the pots; when the soil was highly nutritious and homogeneous, more root biomass was placed in these locations (Figure 3). The plants more readily utilized the soil they had first access to when not facing direct competition for a patch equally accessible to both plants, and the soil was fertile. This foraging strategy would reduce the need to extensively search the rest of the pot for a potential nutrient patch or higher-quality soil. Plants experience a trade-off between exploration of the environment and exploitation of resources [106,107], which is comparable to animals as they move and forage across landscapes [108,109]. Thus, they may invest more energy in pre-empting resources within a high-nutrient patch, ensuring direct access by increasing preliminary root growth [49,100].

### 3.4. Future Directions

Crop systems are drastically different from most natural biological systems, precisely due to their organization in closely planted monoculture fields to promote maximum yield from minimum space on the landscape. The dynamics of plant–plant interactions in these synthetically constructed plant communities have been prone to drastic change through intentional and unintentional artificial selection, especially in cereal crops such as wheat [18,110]. A new avenue of exploration could involve looking at past crop selection and the extent to which plants select for or against kin. Future studies could apply an evolutionary lens by comparing wild ancestors and domesticated cultivars.

Additionally, most research on kin/non-kin effects on spatial distribution and fitness has been genetically limited. Most studies look at half- to full-siblings versus strangers without quantification or scale of genetic relatedness. Further testing would be needed to fill this gap in the literature and determine the genetic distance required for a kin/stranger effect within wheat. These studies would be especially pertinent since kin selection can lead to better group fitness outcomes directly tied to increases in grain production [57,81,83,111]. These findings may have important implications for ecosystem functioning and agriculture. We must understand the underlying mechanisms better to apply this knowledge and enhance crop performance effectively.

## 4. Materials and Methods

### 4.1. Study Species

We chose five cultivars of Canada Western Red Spring (CWRS) wheat: Lillian, Glenn, Carberry, C.D.C. Titanium, and Go Early. Lillian is a solid-stemmed cultivar with high protein content and resistance to wheat stem sawfly and stripe rust [69,112]. Glenn also has high protein content, with additional resistance to Fusarium head blight, leaf rust, and stem rust [69,113]. Carberry is high-yielding with leaf rust resistance [69,70,114,115], while CDC Titanium is midge-tolerant with resistance to Fusarium head blight and leaf and stripe rusts [69,116]. Finally, as the name suggests, Go Early matures early and produces a high yield with common bunt and rust resistance [69,117]. We attained seeds for all cultivars from the Spaner Research Lab at the University of Alberta in Edmonton, AB.

### 4.2. Experimental Design Overview

In brief, we created experimental mesocosms in round 15.2 cm plastic pots with a volume of 1.75 L filled with low-nutrient soil [118,119,120,121] (Figure 4). We planted one or two of the same or different wheat varieties in each pot. Each pot received 1 of 4 soil nutrient treatments (high vs. low fertility × heterogeneous vs. homogeneous distribution), resulting in 90 treatment combinations. Pots were organized into 5 replicate blocks, each containing 1 or 3 replicates of each treatment combination (3 replicates if plants were grown individually; 1 replicate if 2 plants were in each pot). In total, there were 600 experimental pots [(15 pairwise combinations × 4 soil treatments × 5 blocks) + (5 cultivars × 3 replicates × 4 soil treatments × 5 blocks)], all grown on the roof of the University of Alberta Biotron in a randomized block design.

### 4.3. Soil Treatments, Neighbour Treatments, and Plant Growth

The soil in all pots consisted of a low-nutrient soil mix (3:1 sand-to-topsoil mixture (Canar Rock Products Ltd., Edmonton, Canada)), which has been used in other root foraging experiments within the lab group [14,122]. We added 2 levels (0.551 g/L and 4.403 g/L) of slow-release 14:14:14 NPK fertilizer to create low- and high-soil-fertility treatments. For homogenous treatments, the fertilizer was mixed evenly throughout the soil. The fertilizer was placed in a 1 cm patch spanning the entire pot depth in the heterogeneous treatments. When present, nutrient patches were placed equidistant from both plants (Figure 4). The locations of the soil patches in the single and neighbour treatments were the same. Additionally, the total nutrients added per pot were identical between the homogenous and heterogenous treatments within a soil fertility treatment, either high or low.

Three seeds of a given genotype were planted at each location, and seedlings were then thinned to one individual per location within three days, retaining the largest individual. Treatments comprising two plants had each plant placed halfway between the edge and centre of the pot (Figure 4). We planted the seeds in the same pattern for the plants grown alone, with one of the planting locations in the pot remaining empty (Figure 4). The plants received natural sunlight, and we watered them throughout their growth.

### 4.4. Harvest

The plants grew for an average of 43 days, from 7 June to 20 July 2018. We harvested the plants after the seeds had developed to capture maximum reproductive growth. However, to prevent plants from outgrowing the pots below ground and altering root distribution, we harvested the plants before the seeds were fully ripened. To harvest, we clipped each plant at the soil surface. We separated the biomass into two categories: growth towards or away from the centre of the pot relative to the initial planting location (Figure 4). We used this separation to quantify aboveground shifts from a neighbour’s presence. These 2 categories were split into reproductive and non-reproductive biomass for each plant, dried at 65 °C for a minimum of 48 h, and weighed.

According to our research questions, we took five root cores per pot in specified locations using steel root corers (Figure 4). First, to determine patch use, we harvested roots from where the nutrient patch was placed in heterogeneous treatments (Figure 4, core 2). To assess preferential patch use, we also harvested root biomass in the nutrient-empty space directly opposite the patches (Figure 4, core 4). Then, to investigate the use of shared soil, we took soil cores in the central shared soil space (Figure 4, core 3) as well as the spaces each plant had singular access to, between the plants and the pot edges (Figure 4, cores 1 and 5). Finally, we collected all remaining root fragments in the pots. Each root core was gently washed over a 1 mm sieve, removing the soil. After extracting the root fragments using tweezers, we dried them in a drying oven at 65 °C for 48 h prior to weighing them. However, due to a lack of visual differentiation between the roots of the wheat plants, especially those with kin neighbours of the same genotype, cultivar determination and separation were impossible for the cores or root fragments.

### 4.5. Statistical Analysis

To analyse our data, we ran a priori contrasts between the nutrient level and nutrient homogeneity and the overall effect of neighbouring plants, allowing us to compare differences between plants grown with a neighbour or without. We also ran a priori contrasts looking at the effects of kin versus strangers, disregarding single plants. We created beta regression models using the glmmTMB package [123] in R [124] to assess biomass allocation, fitness metrics, and root proliferation, with blocks as random effects. In each of the four beta regression models, the individual effects of neighbours, soil fertility, and nutrient distribution were modelled. Additionally, the two-way interactions between neighbour identity and fertility, neighbour identity and distribution, and soil fertility and nutrient distribution were modelled. Finally, the effects of the three-way interaction between neighbour identity, soil fertility, and nutrient distribution were also analysed. We also ran planned contrasts with neighbour identity and nutrient treatments to determine their impacts on reproductive effort, shoot and root asymmetry, and root precision. For neighbour contrasts, results were averaged over the levels of nutrient treatments, while for nutrient treatments, results were averaged over the levels of kin.

We calculated reproductive effort by taking the proportion of biomass containing reproductive structures and dividing it by the aboveground biomass for each plant. Aboveground biomass asymmetry was measured as the proportion of shoots, including both reproductive and non-reproductive parts, of each plant that grew towards the shared space in the centre of the pot compared with the biomass in shared space that grew away from the centre. Similarly, to study root asymmetry, we measured the proportion of root biomass in the shared soil space (Figure 4, core 3) and compared it with the overall root biomass found in both shared and independent space (Figure 4, cores 1, 3, and 5). To prevent dividing by 0 when there was no biomass growing away from the centre, we added 0.005 g to all biomass values. Finally, to examine root foraging precision, we took the proportion of roots harvested in the patch location (Figure 4, core 2) and divided it by the root biomass harvested in the patch and an equidistant location without a nutrient patch within the pot (Figure 4, cores 2 and 4), creating a pot-level measure. For the belowground measures, we utilized a pot-level measure since visual differentiation between fragmented roots from two neighbouring wheat plants was impossible, preventing investigation of individual space use.

## Figures and Tables

**Figure 1 plants-12-02527-f001:**
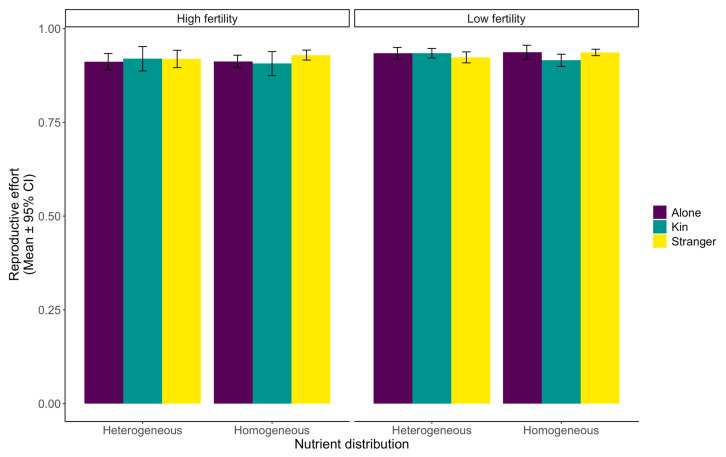
Wheat (*Triticum aestivum*) reproductive effort (mean ± 95% confidence interval) in response to identity, nutrient distribution, and fertility. The reproductive effort was measured as the amount of reproductive biomass the plant grew divided by the plant’s total aboveground biomass.

**Figure 2 plants-12-02527-f002:**
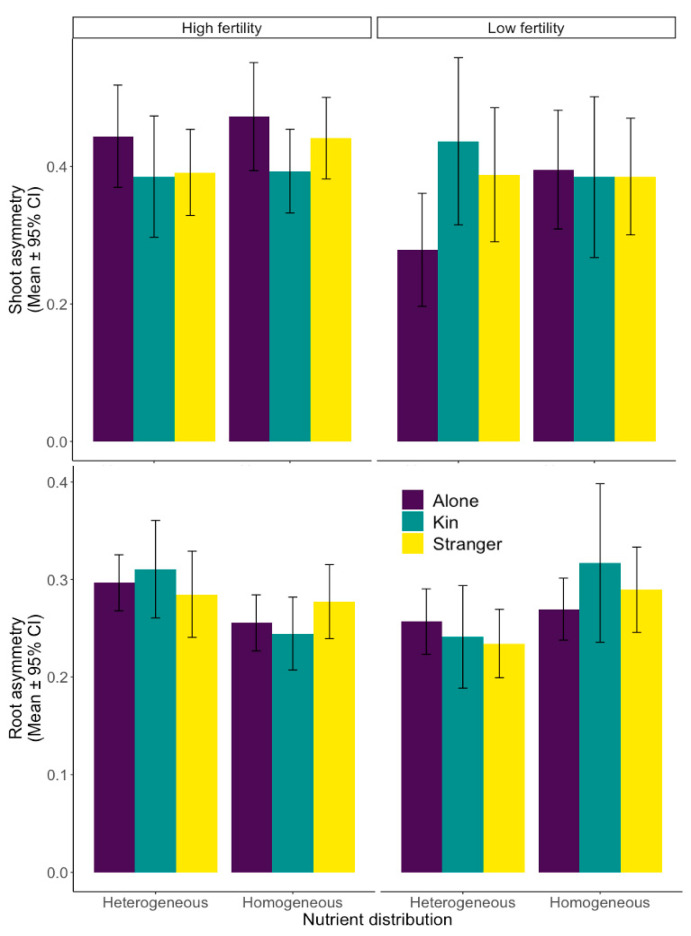
Wheat (*Triticum aestivum*) shoot and root asymmetry (mean ± 95% confidence interval) in response to identity, nutrient distribution, and fertility. Shoot asymmetry is the proportion of each plant’s shoot biomass grown toward the centre of the pot and a neighbour divided by the plant’s overall shoot biomass grown. The root asymmetry is the proportion of roots harvested in the central shared soil space divided by the total roots harvested in the central shared space and the soil between each plant and the pot edge.

**Figure 3 plants-12-02527-f003:**
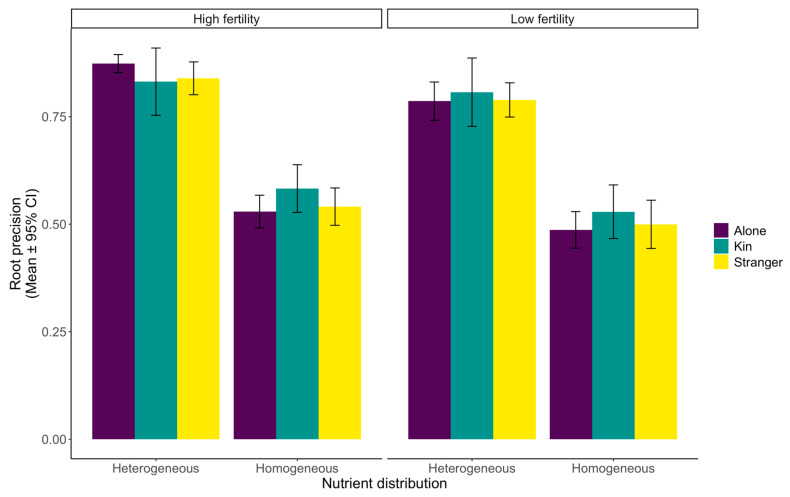
Wheat (*Triticum aestivum*) root precision (mean ± 95% confidence interval) in response to identity, nutrient distribution, and fertility. Precision was measured as the proportion of biomass harvested where the nutrient patch was in the pot divided by the total biomass harvested from the patch and an area equidistant to the plants without a nutrient patch present. This created a pot-level measure.

**Figure 4 plants-12-02527-f004:**
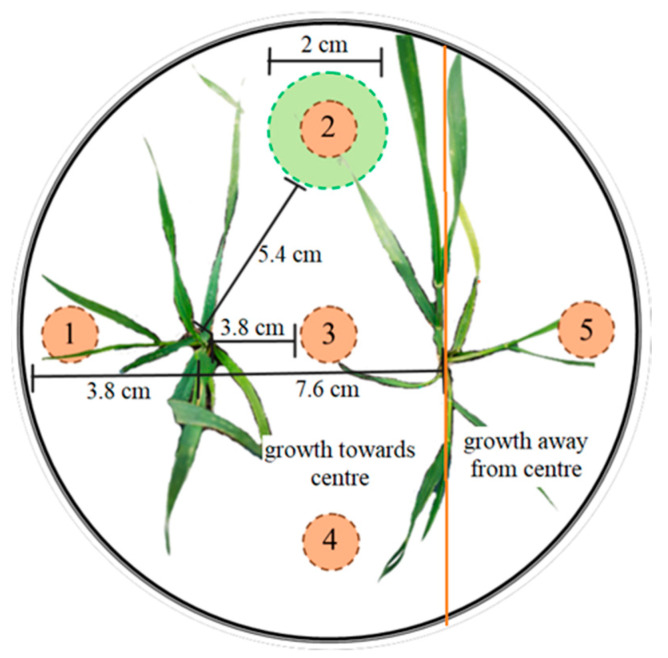
Planting design of wheat, *Triticum aestivum* L., in heterogenous and homogenous soil treatments. The distances between the plants (7.6 cm), the plants and pot edge (3.8 cm), and the plants and centre of the pot (3.8 cm) are displayed. Control treatments had a single plant placed in one of these two locations. The nutrient patch, depicted by the green circle, was placed equidistant from both plants, and the diameter of the patch (2 cm) and its distance from the plants (5.4 cm) are illustrated. At harvest, shoot biomass was separated into growth towards or away from the pot’s centre, as shown by the orange line. We also took five root cores, illustrated by the numbered brown circles.

**Table 1 plants-12-02527-t001:** The effects of neighbour identity, soil fertility, and nutrient distribution on shoot and root asymmetry, root precision, and reproductive effort. The results are from 4 beta regressions, and bolded values represent significant relationships (alpha = 0.05).

Model	F-Value (df1, df2)	*p*-Value
Reproductive effort		
neighbour	2.518 (2, 580)	0.0815
fertility	4.494 (1, 580)	**0.0344**
distribution	1.005 (1, 580)	0.3166
neighbour:fertility	1.938 (2, 580)	0.1449
neighbour:distribution	0.979 (2, 580)	0.3763
fertility:distribution	0.100 (1, 580)	0.7524
neighbour:fertility:distribution	0.331 (2, 580)	0.7185
Shoot asymmetry		
neighbour	1.522 (2, 583)	0.219
fertility	7.243 (1, 583)	**0.0073**
distribution	0.66 (1, 583)	0.4169
neighbour:fertility	2.758 (2, 583)	0.0643
neighbour:distribution	0.617 (2, 583)	0.5397
fertility:distribution	0.071 (1, 583)	0.7905
neighbour:fertility:distribution	0.271 (2, 583)	0.7628
Root asymmetry		
neighbour	0.152 (2, 582)	0.8589
fertility	1.622 (1, 582)	0.2034
distribution	0.538 (1, 582)	0.4634
neighbour:fertility	0.044 (2, 582)	0.9572
neighbour:distribution	1.062 (2, 582)	0.3463
fertility:distribution	11.975 (1, 582)	**0.0006**
neighbour:fertility:distribution	0.878 (2, 582)	0.4163
Root precision		
neighbour	0.634 (2, 582)	0.5311
fertility	10.739 (1, 582)	**0.0011**
distribution	330.789 (1, 582)	**<0.0001**
neighbour:fertility	0.233 (2, 582)	0.7926
neighbour:distribution	0.696 (2, 582)	0.4991
fertility:distribution	0.615 (1, 582)	0.4333
neighbour:fertility:distribution	0.577 (2, 582)	0.562

**Table 2 plants-12-02527-t002:** Planned contrasts with neighbour identity and nutrient treatments for impacts on reproductive effort, shoot and root asymmetry, and root precision. For neighbour contrasts, results are averaged over the levels of nutrient treatments, while for nutrient treatments, results are averaged over the levels of kin. Estimates and standard errors represent log odds ratios, and bolded values represent significant relationships (alpha = 0.05).

Model	Estimate	S.E.	df	t Ratio	*p*-Value
Reproductive effort					
Alone—Neighbour	0.115	0.0573	580	2.003	**0.0456**
Kin—Stranger	−0.115	0.0811	580	−1.413	0.1583
Heterogeneous High fertility—Homogeneous High fertility	0.0796	0.0835	580	0.954	0.3407
Heterogeneous High fertility—Heterogeneous Low fertility	−0.11	0.0875	580	−1.257	0.2094
Homogeneous High fertility—Homogeneous Low fertility	−0.1481	0.0839	580	−1.765	**0.0781**
Heterogeneous Low fertility—Homogeneous Low fertility	0.0415	0.0873	580	0.475	0.6349
Shoot asymmetry					
Alone—Neighbour	−0.1828	0.105	583	−1.733	0.0837
Kin—Stranger	0.0329	0.154	583	0.214	0.8309
Heterogeneous High fertility—Homogeneous High fertility	−0.1221	0.16	583	−0.763	0.4455
Heterogeneous High fertility—Heterogeneous Low fertility	0.2756	0.162	583	1.703	0.0891
Homogeneous High fertility—Homogeneous Low fertility	0.3358	0.159	583	2.113	**0.035**
Heterogeneous Low fertility—Homogeneous Low fertility	−0.0619	0.16	583	−0.386	0.6995
Root asymmetry					
Alone—Neighbour	−0.0166	0.0538	582	−0.309	0.7571
Kin—Stranger	0.0401	0.0777	582	0.515	0.6064
Heterogeneous High fertility—Homogeneous High fertility	0.157	0.0803	582	1.952	0.0514
Heterogeneous High fertility—Heterogeneous Low fertility	0.272	0.0824	582	3.301	**0.001**
Homogeneous High fertility—Homogeneous Low fertility	−0.126	0.0801	582	−1.57	0.1169
Heterogeneous Low fertility—Homogeneous Low fertility	−0.241	0.0822	582	−2.931	**0.0035**
Root precision					
Alone—Neighbour	−0.0397	0.0664	582	−0.599	0.5497
Kin—Stranger	0.1027	0.096	582	1.069	0.2853
Heterogeneous High fertility—Homogeneous High fertility	1.372	0.1022	582	13.427	**<0.0001**
Heterogeneous High fertility—Heterogeneous Low fertility	0.288	0.1078	582	2.675	**0.0077**
Homogeneous High fertility—Homogeneous Low fertility	0.177	0.0924	582	1.917	0.0557
Heterogeneous Low fertility—Homogeneous Low fertility	1.261	0.1005	582	12.55	**<0.0001**

## Data Availability

The data presented in this study are available upon reasonable request from the corresponding author.

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
