# Peer review of "Changes in the Amount and Distribution of Soil Nutrients and Neighbours Have Differential Impacts on Root and Shoot Architecture in Wheat (Triticum aestivum)"

_plants, 2023, doi:10.3390/plants12132527_

Round 1
Reviewer 1 Report
The paper is well written and the methodology sound. I have one important criticism that the authors should consider. Wheat is not in any sense a natural plant. It has been bred for one purpose for thousands of years; reproducible yield. The Broadbalk experiments performed at Rothamsted UK in the 19th century reported it disappeared in one year if the field was left fallow; i.e. it lacks competitive ability against weeds and other plants and it has been bred for stability ONLY in crop fields. Therefore the conclusions of this paper only apply to wheat. Please adjust the last sentence in the abstract to make that clear.
Author Response
Point 1: Thank you for your work. With all the changes and modifications, I find the manuscript to be suitable for publication.
Response 1: Thank you for your review and comments! I appreciate your time and energy in editing this manuscript.
Reviewer 2 Report
Competition between plants has a significant impact on the production of crop species, which is mostly studied in terms of crop-weed ratios. Therefore the complexity of this work is of great importance. However, there are some concerns that need to be addressed.
The material and methodology need to be revised, among other things, before the results can be evaluated.
The abstract does not discuss methodology or the theoretical underpinning of the work. It is necessary to rewrite it in such a way that each line of thought is balanced.
The numbers in Figure 4 have no explanation.
Since it is a very complex, multi-factorial experiment, a graphical abstract is needed, because it is difficult to interpret the experimental setup as it stands
Why was this pot measure chosen?
What was the volume of the soil in the pot?
Was this volume sufficient for the rooting process?
On the basis of what literature and studies was this pot size chosen? Insert the reference in the text.
How was reproductive effort determined in Fig 1? It needs to be detailed in the materials and methods
How was root and shoot assymmetry determined in Fig 2? It needs to be detailed in the materials and methods
Fig 2: the format of the figure is not correct, the figure captions are in the error bar.
How was root precision calculated in Fig 3? It needs to be detailed in the materials and methods
Lines 365-366 "These roots were compared to those we harvested in the space directly opposite the patches (Figure 4, core 4)." How exactly were the roots compared? It needs to be detailed in the materials and methods
Lines 367-369 "Then, to investigate the use of shared soil, we took soil cores in the central shared soil space (Figure 4, core 3) as well as the spaces each plant had singular access to, between the plants and the pot edges (Figure 4, cores 1 and 5)." After taking the sample, how did you process/evaluate it. It needs to be detailed in the materials and methods
Author Response
Thank you for your comments! Please see attachment.

Reviewer 3 Report
Plants exhibit differential behaviours through changes in biomass development and distribution in response to environmental cues, which may have far-reaching consequences for crops. The authors conducted a mesocosm experiment to determine root and shoot behavioural responses of wheat, T. aestivum, when grown in homogeneous or heterogeneous, heavily or lightly fertilized soil, and alone or with a neighbour of the same or different genetic identity. The authors' research has certain value, but there are the following problem.
1、Please describe the results of statistical analysis, such as a detailed description of the established regression model.
2、In section of 4.5,do not forget cite R software.
3、The structure of the paper is not standardized, Please adjust the structure of the paper.
4、The paper lacks a conclusion section.
I consider that the English language needs to be modified appropriately.
Round 2
Reviewer 2 Report
Thank you for your work. With all the changes and modifications, I find the manuscript to be suitable for publication.
Author Response

(The authors gave the same response as above.)

Reviewer 3 Report
Although the structure of the paper is inconsistent with that of a normal academic paper, I consider that the paper has certain academic value. It is suggested that the author should modify the structure of the paper to facilitate readers' reading.
I consider that the moderate editing of English language is necessary.
Author Response
Point 1: Although the structure of the paper is inconsistent with that of a normal academic paper, I consider that the paper has certain academic value. It is suggested that the author should modify the structure of the paper to facilitate readers' reading.
Response 1: Thank you for all of your feedback and guidance. With regards to the structure of the paper, though I aim to maximize clarity and readability of my work, I may be unable to alter the paper's structure substantially. The sections are currently in the order outlined by the journal. More clarity on this request would be quite helpful! I sincerely appreciate the time and energy you've taken to edit this manuscript.